# FLOWS ON CONVEX POLYTOPES

## ABSTRACT

We present a framework for modeling complex, high-dimensional distributions on convex polytopes by leveraging recent advances in discrete and continuous normalizing flows on Riemannian manifolds. We show that any full-dimensional polytope is homeomorphic to a unit ball, and our approach harnesses flows defined on the ball, mapping them back to the original polytope. Furthermore, we introduce a strategy to construct flows when only the vertex representation of a polytope is available, employing unique barycentric coordinates. Our experiments take inspiration from applications in constraint based metabolic modeling and demonstrate that our methods approximate exact sampling distributions and achieve fast training and inference times.

## 1 INTRODUCTION

Recent breakthroughs in flow-based generative modeling have propelled normalizing flows (Papamakarios et al., 2019) into the forefront of probabilistic machine learning. Continuous normalizing flows (CNFs) (Chen et al., 2018; Grathwohl et al., 2018) have transformed the way we model complex, high-dimensional distributions by leveraging ordinary differential equations (ODEs). These approaches have been extended to model distributions on Riemannian manifolds (Mathieu & Nickel, 2020), while flow matching techniques (Lipman et al., 2023) have obviated the need to simulate ODEs during training, which too have been extended to Riemannian manifolds (Chen & Lipman, 2024).

Convex polytopes represent an important class of Riemannian manifolds that naturally arise in diverse applications. These include metabolic models (Orth et al., 2010), power systems (Venzke et al., 2021), and portfolio theory (Bachelard et al., 2023). For example, in constraint-based models of metabolism, the metabolic network is constrained to a convex polytope determined by stoichiometric and boundary conditions. Complex distributions over such polytopes often arise in $^{13}$C metabolic flux analysis (MFA) (Wiechert, 2001; Antoniewicz et al., 2007), where one infers biochemical reaction rates from isotopic labeling data, typically measured by mass-spectrometry. In this paper we show how to model complex distributions on high-dimensional polytopes using normalizing flows, opening up the possibility for their application in simulation based inference (Cranmer et al., 2020) for $^{13}$C-MFA.

**Contributions.** We introduce two strategies to model distributions on convex polytopes given either of their two primary representations: the half-space (H-) representation, defined by intersections of linear inequalities, and the vertex (V-) representation, defined by the convex hull of a finite set of points. First, we demonstrate that any convex polytope is homeomorphic to a unit ball and that a mapping can be found given its H-representation. Building upon this insight, we extend recent circular spline flows (Rezende et al., 2020), originally developed for sphere-based transformations, to design flexible flow-based models over the ball and, by extension, the convex polytope. We then show that the ball transformation is also useful for modeling distributions using Riemannian continuous flows. Furthermore, we introduce a novel strategy for constructing normalizing flows when only the vertex (V-)representation is available, relying on unique barycentric coordinates. This strategy effectively bypasses the computational bottleneck associated with converting between polytope representations. Together, these contributions offer a unified and efficient framework modeling distributions on convex polytopes, significantly broadening the scope of flow-based models in real-world applications.

## 2 PRELIMINARIES

Every polytope, denoted $\mathcal{F}$, in this text is implied to be convex without mention. The half-space or H-representation of a polytope $\mathcal{F}^{\ddagger}$ is given by

$$\boldsymbol{S}\,v = h, \quad \boldsymbol{S} \in \mathbb{R}^{M \times R} \tag{1}$$

$$\boldsymbol{C}\,v \leq d, \quad \boldsymbol{A} \in \mathbb{R}^{C \times R} \tag{2}$$

$$\boldsymbol{A}^{\ddagger} = \begin{bmatrix} \boldsymbol{S} \\ -\boldsymbol{S} \\ \boldsymbol{C} \end{bmatrix}, \quad b^{\ddagger} = \begin{bmatrix} h \\ -h \\ d \end{bmatrix} \tag{3}$$

$$\mathcal{F}^{\ddagger} = \{v^{\ddagger} \in \mathbb{R}^R \mid \boldsymbol{A}^{\ddagger}\,v \leq b^{\ddagger}\}. \tag{4}$$

equation 1 shows the equality constraints, equation 2 shows the inequality constraints and equation 3 is the canonical H-representation of a polytope; matrices are shown in bold-face. Equivalently, this polytope can be represented in vertex or V-representation as follows

$$\mathcal{F}^{\ddagger} = \{v^{\ddagger} \in \mathbb{R}^R \mid v^{\ddagger} = \boldsymbol{V}^{\ddagger}\,\lambda\,\forall\,\lambda \in \Delta_1^V\}. \tag{5}$$

Matrix $\boldsymbol{V}^{\ddagger} \in \mathbb{R}^{R \times V+1}$ is a matrix whose columns are the extreme points or vertices of the polytope. $\Delta_1^V$ is the $V$ dimensional probability simplex embedded in $\mathbb{R}^{V+1}$, meaning that $\lambda_i \geq 0\,\forall i \in \{1 : V+1\}$ and $\|\lambda\|_1 = 1$. In our notation, subscripts denote elements of vectors or matrices while superscripts are used for disambiguation of objects rather than exponentiation.

Which polytope representation one has access to is determined by the application. Converting from the V to the H representation of a polytope has a time-complexity of $O((V+1)^{\lfloor \frac{K}{2} \rfloor})$ (Avis & Fukuda, 1992), where $K \leq R$ is the dimensionality of the polytope, and $R$ is the dimension of the ambient space in which the polytope is embedded. Converting between V and H representations is only computationally feasible for low-dimensional polytopes with few vertices.

### 2.1 DISCRETE AND CONTINUOUS NORMALIZING FLOWS ON RIEMANNIAN MANIFOLDS

Discrete normalizing flows transform a simple base density $q^0(Y^0)$ via a composition of diffeomorphisms $f = f^L \circ \cdots \circ f^1$ into a more complex one. The diffeomorphisms are functions with learnable parameters. Upon training, the complex distribution should resemble the true data distribution, from which data is sampled $y \sim p(Y)$.

Henceforth, manifolds, denoted $\mathcal{M}$, are implied to be Riemannian manifolds without mention. A diffeomorphism $f^i : \mathcal{M}^{i-1} \to \mathcal{M}^i$, $\mathcal{M}^{i-1} \subset \mathbb{R}^Q$ and $\mathcal{M}^i \subset \mathbb{R}^R$ maps between two $K$-dimensional manifolds, both embedded in an ambient space with $R, Q \geq K$. The density of the flow updates according to the change-of-variables formula

$$q^i(y^i) = \frac{q^{i-1}(y^{i-1})}{\det \boldsymbol{G}^i}, \quad y^i = f^i(y^{i-1}), \quad \boldsymbol{G}^i = (\boldsymbol{J}^i \boldsymbol{F}(y^{i-1}))^T \boldsymbol{J}^i \boldsymbol{F}(y^{i-1}), \tag{6}$$

where $\boldsymbol{J}^i \in \mathbb{R}^{R \times Q}$ is the Jacobian of function $f^i$. $\boldsymbol{F}(y^{i-1}) \in \mathbb{R}^{Q \times K}$ is an othonormal frame of the tangent space $T_{y^{i-1}}\mathcal{M}$ at point $y^{i-1} \in \mathcal{M}^{i-1}$. See appendix A of (Rezende et al., 2020) for more details. The Jacobian of function $f^i$ is

$$\boldsymbol{J}^i = \left[\frac{\partial y_j^i}{\partial y_k^{i-1}}\right]_{j,k}, \boldsymbol{J}^i \in \mathbb{R}^{Q \times R}. \tag{7}$$

In the case that $R = Q = K$, and where $\mathcal{M}^i$ and $\mathcal{M}^{i-1}$ are diffeomorphic, matrix $\boldsymbol{F}(y^{i-1})$ is orthogonal and the density update reduces to $q^i(y^i) = q^{i-1}(y^{i-1})|\det \boldsymbol{J}^i|^{-1}$. For discrete flows, triangular Jacobians are preferred for computational efficiency. A host of such transformations and their respective trade-offs is described in (Papamakarios et al., 2019). The total density update of a flow composed of several diffeomorphisms is given by $q(y^L) = q(y^0) \prod_{i=1}^L (\det \boldsymbol{G}^i)^{-1}$.

In the continuous limit, as the number of composed flows tends to infinity, the discrete sequence of transformations is replaced by a continuous evolution governed by an ordinary differential equation

(ODE). This leads to continuous normalizing flows (CNF) or, equivalently, neural ODEs (Chen et al., 2019; Grathwohl et al., 2018) which too have been extended to Riemannian manifolds (Mathieu & Nickel, 2020). In continuous normalizing flows on a Riemannian manifold $\mathcal{M}$, one defines a flow $\psi : \mathcal{M} \times [0,1] \to \mathcal{M}$ as the solution of the ODE

$$\frac{d\psi_t}{dt} = u^t(\psi_t, t; \theta), \text{ with initial condition } \psi_0 \sim q_0(\Psi_0) \tag{8}$$

where $u^t : [0,1] \times \mathcal{M} \to T\mathcal{M}$ is a timedependent vector field that is parametrized by a neural network. $q_0(\Psi_0)$ is a simple base density over the manifold and we denote $\psi^t = \psi(y, t)$ with time $t \in [0,1]$. Pushing forward a base density $q_0$ through $\psi_t$ yields a probability path

$$q_t(y) = q_0(\psi_t^{-1}(y)) \cdot \exp^{-1}\Big(\int_0^t \nabla_g u_t(x_s)ds\Big), \quad x_s = \psi_s(\psi_t^{-1}(y)) \tag{9}$$

Where $\nabla_g u_t$ is the Riemannian divergence. Flow matching (Lipman et al., 2023; 2024) avoids ODE integration during training by aligning $u^t$ with a target vector field $u^*$. This approach has also been extended to Riemannian manifolds (Chen & Lipman, 2024), where the target vector field is chosen to be geodesic

$$u^*(t) = \exp_{\psi_0}(t \ln_{\psi_0}(\psi_1)) \tag{10}$$

where $\exp$ is the exponential map and $\ln$ is the logarithmic map. In practice, the neural network used to parametrize $u^t$ is defined in the ambient space of the manifold and a projection operator

$$\pi(x) = \arg\min_{y \in \mathcal{M}} \|y - x\|_g \tag{11}$$

is used to project $x$ to the tangent space of the manifold at $y$.

## 3 FLOWS ON BALLS AND BARYCENTRIC COORDINATES

In most applications, we are given the H-representation of a $K$-dimensional polytope, which is commonly embedded in some ambient space $\mathbb{R}^R$. We begin by showing how to transform such a polytope into its corresponding full-dimensional John polytope, which is centered at the origin and whose facets all touch the unit ball $\mathbb{B}^K(1)$. We subsequently map the John polytope to the unit ball. The first step consists of determining the minimal affine subspace. Every point $v^\ddagger \in \mathcal{F}^\ddagger$ can be expressed in terms of free variables $v^\dagger \in \mathbb{R}^K$ via an affine embedding

$$v^\ddagger = \boldsymbol{T} v^\dagger + \tau, \quad \boldsymbol{T} \in \mathbb{R}^{R \times K}, \quad K \leq R. \tag{12}$$

where $\boldsymbol{T}$ and $\tau$ are determined by removing redundant constraints through solving linear programs for each inequality and by collecting implicit equalities from $\boldsymbol{C}$ into an extended equality constraint matrix $\boldsymbol{S}^+$. Section 2.2.1 in (Liphardt, 2018) offers a detailed and graphical explanation of these steps. In practice, two embedding strategies are employed. We define the Chebyshev center as

$$v^0 = \arg\min_{v^0} \max_{v^\ddagger \in \mathcal{F}^\ddagger} \|v^\ddagger - v^0\|_2^2, \tag{13}$$

and then determine the row reduced echelon form (RREF) embedding by setting

$$\boldsymbol{T}^{\text{rref}} = \frac{\partial v^\ddagger}{\partial v^\dagger} = \begin{bmatrix} \boldsymbol{I} \\ \boldsymbol{D}^\star \end{bmatrix} \quad \text{s.t.} \quad v^\ddagger = \begin{bmatrix} v^\dagger \\ v^\star \end{bmatrix}, \quad \tau^{\text{rref}} = v^0 - \boldsymbol{T}^{\text{rref}} v^0_{:K}, \tag{14}$$

where $:K$ denote the first $K$ elements of a vector, $\boldsymbol{I}$ is the identity matrix and $v^*$ denote the dependent variables. Alternatively, the singular value decomposition (SVD) of $S^+$ can be used as the embedding

$$S^+ = \boldsymbol{U} \boldsymbol{\Sigma} \boldsymbol{V}^T, \quad \boldsymbol{T}^{\text{SVD}} = \boldsymbol{V}_{:,-K:}, \quad \tau^{\text{SVD}} = v^0. \tag{15}$$

Using either embedding, the full-dimensional polytope in free-variable space is defined as

$$\mathcal{F}^\dagger = \{v^\dagger \in \mathbb{R}^K \mid \boldsymbol{A}^\dagger v^\dagger \leq b^\dagger\}, \tag{16}$$

$$\boldsymbol{A}^\dagger = \boldsymbol{A}^\ddagger \boldsymbol{T}, \quad b^\dagger = b^\ddagger - \boldsymbol{A}^\ddagger \tau. \tag{17}$$

In $^{13}$C-MFA the RREF embedding is preferred because the free variables are just a subset of the original fluxes (Quek et al., 2009), while the SVD embedding is more general (Haraldsdóttir et al., 2017; Theorell et al., 2022).

To facilitate the modeling of distributions, the transformed polytope is further "rounded" to a maximum isotropic (John) position (John, 2014). An ellipsoid centered at $\epsilon$ is defined by

$$\mathcal{E} = \left\{ v^\dagger \in \mathbb{R}^K \,\Big|\, (v^\dagger - \epsilon)^T \, (\boldsymbol{E}\,\boldsymbol{E}^T)^{-1} \, (v^\dagger - \epsilon) \leq 1 \right\}, \tag{18}$$

so that the free variables $v^\dagger$ can be written in terms of rounded variables $v$ as

$$v^\dagger = \boldsymbol{E}\,v + \epsilon. \tag{19}$$

Finding the maximum volume ellipsoid (MVE) contained in $\mathcal{F}^\dagger$ is a convex optimization problem that maximizes the determinant of $\boldsymbol{E}$ and that can be solved efficiently (Zhang & Gao, 2003). Using this affine transformation, the John polytope is given by

$$\mathcal{F} = \{v \in \mathbb{R}^K \mid \boldsymbol{A}\,v \leq b\}, \tag{20}$$

$$\boldsymbol{A} = \boldsymbol{A}^\dagger \boldsymbol{E}, \quad b = b^\dagger - \boldsymbol{A}^\dagger \epsilon, \tag{21}$$

such that $\mathbb{B}^K(1) \subseteq \mathcal{F} \subseteq \mathbb{B}^K(\Phi)$ holding for some radius $\Phi > 1$.

The next step is to map the John polytope $\mathcal{F}$ to the unit ball $\mathbb{B}^K(1)$. This transformation can be thought of as the inverse of the hit-and-run (HR) transform (Smith, 1984; Kaufman & Smith, 1998) employed in the uniform sampling of polytopes. For a given point $v \in \mathcal{F}$, we first compute its Euclidean norm

$$d = \|v\|_2, \tag{22}$$

and normalize its direction

$$s = \frac{v}{d}, \quad s \in \mathbb{S}^{K-1}(1). \tag{23}$$

The distances to all hyperplanes are then computed via

$$\alpha = b \oslash (\boldsymbol{A}\,s), \tag{24}$$

and the minimal positive distance is given by

$$\alpha^{\min} = \min\{\alpha \mid \alpha \geq 0\}. \tag{25}$$

Next, the $K$-norm is defined by

$$r = \left(\frac{d}{\alpha^{\min}}\right)^{\frac{1}{K}}, \quad r \in [0, 1], \tag{26}$$

and the ball coordinates are obtained as

$$\beta = r \cdot s, \quad \beta \in \mathbb{B}^K(1). \tag{27}$$

This entire mapping defines the homeomorphism

$$\mathcal{b} : \mathcal{F} \to \mathbb{B}^K(1), \qquad v \mapsto \beta, \tag{28}$$

$$\mathcal{b}^{-1} : \mathbb{B}^K(1) \to \mathcal{F}, \qquad \beta \mapsto v = \alpha^{\min}\,\beta.$$

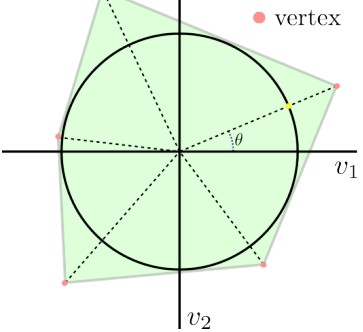

Figure 1: *Graphical intuition for the homeomorphism between a 2D convex polytope and the disk.*

An intuition for mapping $\mathcal{b}$ is shown in figure 1. The green area represents a polytope in John position. For a chord between the origin and a face of the polytope, mapping $\mathcal{b}$ squishes it such that it comes to lie within the disk.

Along directions where a chord points towards a $K - n$-face where $K \geq n > 1$, e.g. a vertex (0-face) or an edge (1-face), the $\min$ function of equation 25 is not defined, since multiple hyper-planes are at exactly equal distance. This yields a discontinuity in the derivative. In figure 1 the directions along which no derivative is defined are shown as dotted lines. The set of points for which the derivative is not defined has measure 0, therefore the discontinuity will not matter in practice.

### 3.1 DISCRETE AND CONTINUOUS FLOWS ON THE BALL AND THE POLYTOPE

For modelling discrete flows on the ball, we adopt spline flows (Durkan et al., 2019b) combined with the recursive cylinder-coordinate parameterization on the sphere introduced in (Rezende et al., 2020) and generalize it to the closed ball $\mathbb{B}^K$ by introducing a radial coordinate $r$. For completeness and to keep the presentation self-contained, we reproduce this construction in Appendix A, where we also provide a full derivation of the corresponding Jacobian.

The ball-homeomorphism turns out to also be useful when modelling distributions using CNFs. One challenge with CNFs is that, during inference, integrating the underlying ODE may yield solutions that leave the manifold. To counteract this, variables are projected back onto the manifold at each integration step using the projection operator defined in equation 11 (Chen & Lipman, 2024). For a polytope equipped with a Eucliean metric, this projection requires solving the following quadratic program

$$\pi(h^t) = \arg\min_{v \in \mathcal{F}} \|h^t - v\|_2^2 \tag{29}$$

$$= \arg\min_{\boldsymbol{A} \cdot v \leq b} \|h^t - v\|_2^2. \tag{30}$$

Solving a quadratic program at every ODE iteration, whenever the solution exits the polytope, is computationally demanding[1].

Euclidean flows (Lipman et al., 2023) are defined on $\mathbb{R}^K$ and use a Euclidean metric. They do not inherently enforce the boundary constraints of a polytope. If both the base and target distributions are confined to a polytope, a perfectly matched Euclidean flow, i.e. one for which $KL(q_\theta^{eucl}\|p(y)) = 0$, should, in principle, never generate probability paths outside the polytope. We therefore investigate Euclidean flows, defined as

$$\begin{cases} q^{eucl}(v) = p_{\mathcal{F}}^{\mathcal{U}}(v) \cdot \exp^{-1}\left(\int_0^1 \nabla u_t(x_s)ds\right) & \text{if } v \in \mathcal{F} \\ 0 & \text{else} \end{cases} \tag{31}$$

where samples that fall outside of the polytope are rejected. In reality, the flow will not exactly match the target distribution. Therefore we need to divide $q^{eucl}(v)$ by normalizing constant $Z_{eucl} = \int_{\mathcal{F}} q^{eucl}(s)ds$ to obtain the normalized density.

If we transform the polytope into a ball and use a CNF to model a distribution on the ball, then the projection operator is a simple scaling operation

$$\pi(h^t) = \frac{y}{\max(1, \|h^t\|_2)}. \tag{32}$$

Though it is possible to include the boundary of the ball in Riemannian flow matching (section G.2 of (Chen & Lipman, 2024)) we restrict our attention to the open ball, which corresponds to the interior of the polytope. The density of a point in the open polytope is given by

$$q^{ball}(v) = p_{\mathbb{B}}^{\mathcal{U}}(\beta) \cdot \exp^{-1}\left(\int_0^1 \nabla_g u_t(x_s)ds\right) \cdot \text{abs}\,|\boldsymbol{J}^{v\beta}|^{-1} \tag{33}$$

where $\boldsymbol{J}^{v\beta}$ is the Jacobian of function $\mathcal{b}^{-1}$, which maps points from the ball to the polytope.

For exact density evaluation, it is also necessary to evaluate the normalized density of the base distribution. For the ball flow, it is trivial to choose a distribution whose density can be evaluated analytically, e.g. the uniform density $p_{\mathbb{B}}^{\mathcal{U}}(v) = (\text{vol}\,\mathbb{B})^{-1}$. Conversely, the uniform density over the polytope is $p_{\mathcal{F}}^{\mathcal{U}}(v) = (\text{vol}\,\mathcal{F})^{-1}$, which for lower dimensions can be computed analytically but for higher dimensions can only be approximated through sampling (Cousins & Vempala, 2016; Emiris & Fisikopoulos, 2013).

---

[1]In our implementation, we are limited by the fact that PyTorch currently does not support parallelized gradient tracking through functions that involve data-dependent control flow, see github issue. We thus cannot monitor the convergence of the quadratic program in parallel and would have to resort to sequentially evaluating each sample in a batch

## 3.2 FLOWS ON BARYCENTRIC COORDINATES

If we only have access to the V-representation of a polytope, we would like to still be able to model distributions over it without having to convert to the H-representation, which might be computationally infeasible. For this Section, we consider a full-dimensional polytope $\mathcal{F}$ in V-representation and assume we have samples from a distribution over this polytope.

The barycentric coordinates, $\lambda$ in equation 5, for a point on a simplex are unique since $V = K$. For general polytopes, it is typically the case that $V \gg K$ which makes the system $\boldsymbol{V} \cdot \lambda = v$ under determined, meaning that the mapping $\mathcal{F} \rightrightarrows \Delta_1^V$ is a set-valued function and thus not bijective. We can therefore not apply the change of variables of equation 6. This can be solved by choosing a unique barycentric coordinate mapping. A computationally feasible choice are the maximum entropy coordinates (mec) (Hormann & Sukumar, 2008) defined as

$$\text{mec} : \mathcal{F} \to \Lambda, \qquad v \mapsto \lambda^{\text{mec}} = \arg \max_{\substack{\lambda \in \Delta_1 \\ \boldsymbol{V}\, \lambda = v}} -\lambda^T \ln(\lambda) \tag{34}$$

$$\text{mec}^{-1} : \Lambda \to \mathcal{F}, \qquad \lambda^{\text{mec}} \mapsto v = \boldsymbol{V}\, \lambda^{\text{mec}}. \tag{35}$$

With

$$\Lambda = \{\lambda^{\text{mec}} = \text{mec}(v) \mid v \in \mathcal{F}\}, \tag{36}$$

being the set of all mec transformed coordinates of every point in the open polytope. For brevity, we will drop the mec superscript from $\lambda$ in what follows. Note that the mec mapping is defined only for points in the open polytopes, because at $n$-faces of the polytope, some elements of the barycentric coordinates become zero, and $\ln(0)$ is undefined.

Equations equation 34 and equation 35 define the mec mapping and its inverse, which is bijective by construction. As demonstrated in (Hormann & Sukumar, 2008), the forward mec mapping is smooth over the open polytope, while its inverse is linear and thus smooth. Since the open $K$-dimensional polytope is a Riemannian manifold and the mec mapping along with its inverse are smooth everywhere, we conclude that the set $\Lambda$ forms a smooth connected $K$-dimensional sub-manifold embedded in the simplex $\Delta_1^V$ (Lee, 2012).

A metric on the simplex is provided by the Aitchison geometry (Aitchison, 1982; Greenacre et al., 2023). In this framework, the isometric log-ratio (ilr) transform (Egozcue et al., 2003) is defined as follows:

$$\text{ilr} : \Delta_1^V \to \mathbb{R}^V, \qquad \lambda \mapsto z = \boldsymbol{H}\, \ln(\lambda) \tag{37}$$

$$\text{ilr}^{-1} : \mathbb{R}^V \to \Delta_1^V, \qquad z \mapsto \lambda = \begin{cases} z^a &= \exp(\boldsymbol{H}^T z) \\ \lambda &= \frac{z^a}{\boldsymbol{1}^T z^a} \end{cases} \tag{38}$$

Here, $\boldsymbol{H} \in \mathbb{R}^{K \times V}$ is taken to be the Helmert matrix, which provides an orthonormal basis for the subspace of $\mathbb{R}^V$. Both the ilr and its inverse are smooth, thus making the combined $\text{mec} \circ \text{ilr}$ mapping a diffeomorphism.

Because the ilr transform is an isometry between the Aitchison geometry on the simplex and Euclidean space, the linear interpolation between any two points in $\mathbb{R}^K$ corresponds to the geodesic, with respect to the Aitchison metric, between the corresponding points in $\Delta_1^V$. An alternative geometric structure on the simplex is provided by the Fisher–Rao metric (Davis et al., 2024), but we did not investigate this option further.

Let

$$\mathcal{Z} = \{z = \text{ilr}(\lambda) \mid \lambda \in \Lambda\} \tag{39}$$

be the ilr-transformed image of $\Lambda$. Since the ilr transform is an isometry, $\mathcal{Z}$ is a $K$-dimensional affine sub-space of $\mathbb{R}^V$. We compute an orthogonal projection using singular value decomposition (SVD). Let the columns of matrix $\boldsymbol{Z}$ represent at least $K$ ilr-transformed points from $\Lambda$, one may then write:

$$\boldsymbol{Z} = \boldsymbol{U}\, \boldsymbol{\Sigma}\, \boldsymbol{W}^T \tag{40}$$

$$\boldsymbol{P} = \boldsymbol{W}_{:,:K}^T \tag{41}$$

Where $\boldsymbol{P}$ equals the first $K$ rows of $\boldsymbol{W}^T$; this matrix can be thought of as a mapping to the orthonormal frame of equation 6. To then obtain the projected ilr points, we define the projection:

$$\text{proj} : \mathbb{R}^V \to \mathbb{R}^K, \qquad\qquad z \mapsto z^p = \boldsymbol{P}\, z \qquad\qquad (42)$$

$$\text{proj}^{-1} : \mathbb{R}^K \to \mathbb{R}^V, \qquad\qquad z^p \mapsto z = \boldsymbol{P}^T z^p \qquad\qquad (43)$$

The projected ilr transformation recovers the effective $K$-dimensional coordinates of the ilr-transformed points. In practice, we first transform all samples from the target distribution into ilr coordinates, and then compute a singular value decomposition (SVD) on a large batch of data to determine an optimal $K$-dimensional projection. We denote the set of projected coordinates as:

$$\mathcal{Z}^p = \{ z^p = \text{proj}(z) \mid z \in \mathbb{R}^V \} \qquad\qquad (44)$$

Once the target coordinates have been mapped to these projected ilr coordinates, we standardize the data by subtracting the mean $\mu$ and dividing by the standard deviation $\sigma$

$$\text{stdz} : \mathbb{R}^K \to \mathbb{R}^K, \qquad\qquad z^p \mapsto z^t = (z^p - \mu) \oslash \sigma \qquad\qquad (45)$$

$$\text{stdz}^{-1} : \mathbb{R}^K \to \mathbb{R}^K, \qquad\qquad z^t \mapsto z^p = z^s \odot \sigma + \mu \qquad\qquad (46)$$

In this way, every sample from the target distribution is represented in projected, standardized ilr coordinates $z^t$, and the full mapping $\text{stdz} \circ \text{proj} \circ \text{ilr} \circ \text{mec} : v \mapsto z^t$ is a diffeomorphism. We can then model their distribution using either discrete flows or CNFs. In our case, we use a Euclidean CNF whose density is:

$$q^{ait}(v) = p_{\mathbb{R}}^{\mathcal{N}}(z^t) \cdot \exp^{-1}\!\left( \int_0^1 \nabla u_t(x_s) ds \right) \cdot |\underbrace{\text{diag}(\sigma)\, \boldsymbol{P}^T \boldsymbol{J}^{ilr} \boldsymbol{V}}_{\boldsymbol{J}^{vt}}|^{-1} \qquad (47)$$

In this expression, $p_{\mathbb{R}}^{\mathcal{N}}(z^t)$ denotes the Gaussian base density. If we track the dimensions through each change of variables, we see that the overall Jacobian, $\boldsymbol{J}^{vt}$, is a square matrix. In particular, we have $\text{diag}(\sigma) \in \mathbb{R}^{K \times K}$, $\boldsymbol{P}^T \in \mathbb{R}^{K \times V}$, $\boldsymbol{J}^{ilr} \in \mathbb{R}^{V \times (V+1)}$ and $\boldsymbol{V} \in \mathbb{R}^{(V+1) \times K}$. The product of these matrices yields $\boldsymbol{J}^{vt} \in \mathbb{R}^{K \times K}$. Its determinant precisely captures the change in density resulting from the entire sequence of transformations.

## 4 EXPERIMENTS

We consider two target densities. The first, $p_{\mathcal{F}}^{mog}$, is a mixture of three Gaussians supported on a $K = 4$ dimensional polytope $\mathcal{F}$ derived from a small metabolic model. The equality and inequality constraints for this metabolic model are described in Appendix Appendix B, which also details the derivation of the John polytope from this metabolic model. In that appendix we also give the full specification the three-component Gaussian mixture (means, weights, covariances), together with the computation of the normalizing constant $Z_{\mathcal{F}}$, which ensures that the density inside the polytope sums to 1.

Our second target, $p_{\square}^{mog}$ is a mixture of Gaussians constrained $K = 20$ hyper-rectangle $\square$ and reuses the same mixture weights and covariances as $p_{\mathcal{F}}^{mog}$, with each mean having a single nonzero entry $1.015$ in one of the first three dimensions; the normalizing constant $Z_{\square}$ is computed analogously. Because the unit ball inscribed in $\mathcal{F}$ or $\square$ touches every facet, choosing $\|\mu^i\|_2 \approx 1$ ensures that a substantial portion of the *unconstrained* mixture lies outside the polytope, which stresses support handling.

We generate training datasets by sampling from both target distributions with our custom multi-proposal, random-direction Hit-and-Run sampler, described in appendix C, which also details the sampler's implementation and reports convergence statistics for both MCMC algorithms. Figure 2A shows a density estimate of samples from the training dataset for the target distribution $p_{\mathcal{F}}^{mog}$, and figure 3A shows the corresponding estimate for $p_{\square}^{mog}$.

In each of our models, cylinder-spline flows $q^{\text{spline}}$, ball-CNF $q^{\text{ball}}$, and Aitchison-CNF $q^{\text{ait}}$, sampling proceeds by chaining a simple base distribution through a series of learnable coordinate transforms combined with a fixed mapping until we recover the original polytope variables $v$. During training, we work entirely in the base-variable space, e.g. $\varphi$ for the spline flow, $\beta$ for the ball CNF, and $z^t$

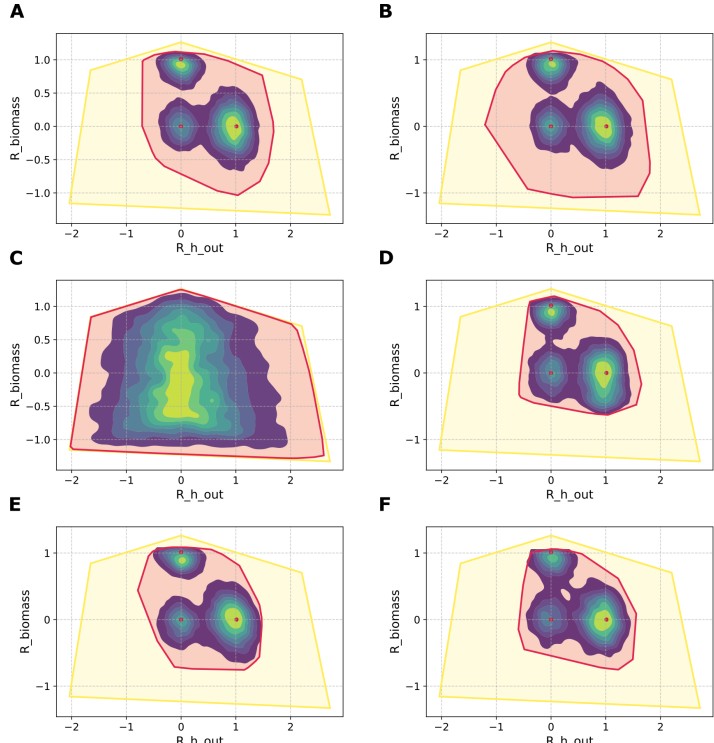

Figure 2: Yellow: the John polytope (projected onto 2D). Red: convex hull of all samples from either the flow or target. Kernel density estimates (from 10k samples) are overlaid, with red points indicating the means of the three Gaussians in the target density $p_{\mathcal{F}}^{mog}$. (A) 105k samples from MCMC; (B) 20k samples from a cylinder spline flow $q^{spline}$; (C) 125k samples from the uniform target $p_{\mathcal{F}}^{\mathcal{U}}$ (via MCMC); (D) 20k samples from a Euclidean CNF $q^{eucl}$; (E) 20k samples from a Riemannian CNF $q^{ball}$; (F) 20k samples from a Euclidean CNF on standardized, projected ilr coordinates $q^{ait}$.

for the Aitchison CNF, which lets us sidestep repeated Jacobian calculations. At inference time, however, we must evaluate the full change-of-variables determinant $\det \mathbf{J}^{v\varphi}$, $\det \mathbf{J}^{v\beta}$, or $\det \mathbf{J}^{vt}$, each of which is a dense $K \times K$ matrix that costs $O(K^3)$ to compute. We noticed that automatic differentiation through these fixed transforms was prohibitively expensive. We therefore derived their Jacobians analytically and evaluate them once per batch in a single matrix-determinant call.

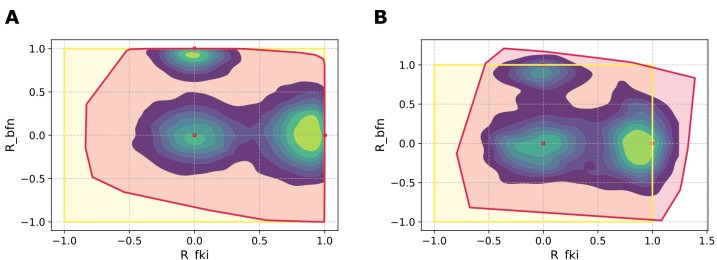

Figure 3: Similar to Figure 2, but here $p_{\square}^{mog}$ is the target. (A) 125k MCMC samples; (C) 20k samples from the Euclidean CNF $q^{eucl}$.

To compare the different flows, we evaluate two key metrics, the effective sample size (ESS), expressed as a percentage of the total samples drawn from the trained flow, and the Kullback-Leibler (KL) divergence between the trained flow and the target distribution. For Euclidean flows, we also report the fraction of generated samples that fall outside the polytope. The precise formulas for these evaluation metrics are given in appendix D, and the results are summarized in Table 1. Figure 2

presents the 2D marginal densities of all flows target $p_{\mathcal{F}}^{mog}$, while figure 3B shows the 2D marginal projection of the Euclidean flow onto the first two dimensions of $p_{\square}^{mog}$.

| Model | target | base | dim $K$ | KL [nats] | ESS (%) | outside (%) |
|---|---|---|---|---|---|---|
| $q^{spline}$ | $p_{\mathcal{F}}^{mog}$ | $p_{\square}^{\mathcal{U}}$ | 4 | 6.747e-01 | 85.7 | - |
| $q^{eucl}$ | $p_{\mathcal{F}}^{mog}$ | $p_{\mathcal{F}}^{\mathcal{U}}$ | 4 | 8.440e-01 | 80.2 | 5.9 |
| $q^{ball}$ | $p_{\mathcal{F}}^{mog}$ | $p_{\mathbb{B}}^{\mathcal{U}}$ | 4 | 8.985e-01 | 63.3 | - |
| $q^{ait}$ | $p_{\mathcal{F}}^{mog}$ | $p_{\mathbb{R}}^{\mathcal{N}}$ | 4 | 5.274e-01 | 79.3 | - |
| $q^{eucl}$ | $p_{\square}^{mog}$ | $p_{\square}^{\mathcal{U}}$ | 20 | 8.236e-02 | 19.8 | 12.5 |

Table 1: KL divergence (nats) w.r.t. target density and effective sample size (ESS) for 20k samples from every trained flow.

For the Aitchison flow, the polytope $\mathcal{F}$ has $V + 1 = 14$ vertices. To verify that the ilr transform correctly recovers the affine subspace corresponding to the image of the manifold $\Lambda$, we compute the singular value decomposition (SVD) of equation 40 on 5000 ilr coordinates. The first five singular values, sorted by magnitude, are:

$$6.3 \times 10^2, \quad 2.7 \times 10^2, \quad 1.9 \times 10^2, \quad 6.3 \times 10^1, \quad 1.4 \times 10^{-13}.$$

This rapid decay confirms that the effective dimension of the data is indeed $K$, and that we can numerically recover the affine subspace for the image of $\Lambda$.

We also compared the model architectures, hyper-parameters, as well as training and inference times for all models. The results of this comparison are displayed in Table 3 in Appendix E. This Appendix elaborates on implementation details and is useful for practitioners that need to make choices based on their needs.

## 5 DISCUSSION

Flows targeting $p_{\mathcal{F}}^{mog}$ show similar overall performance, but important differences arise. Riemannian flow matching guarantees all samples lie within the polytope, while Euclidean flows occasionally produce invalid samples, an issue that worsens in higher dimensions as more mass concentrates near the facets (Figure 3, Table 1). For Riemannian CNFs, frequent quadratic program projections would be required in higher dimensions, motivating the use of ball flows. Although the ball avoids invalid samples, its reduced ESS at comparable KL divergence suggests less accurate tail modeling. These results highlight two challenges for Euclidean CNFs on polytopes: approximate base densities from sampling-based volume estimates and frequent out-of-polytope samples. Transforming to a ball alleviates these difficulties and yields more tractable geometry.

Using the ball as a base manifold also clarifies tradeoffs between CNFs and discrete flows (Table 3). CNFs train nearly an order of magnitude faster due to flow matching, though discrete flows incur delays from autoregressive architectures. Inference times are comparable, with CNFs incurring some overhead from divergence evaluation that could be reduced via estimators such as Hutchinsons trace estimator (Grathwohl et al., 2018). We further introduced flows directly on the V-representation, motivated by high-dimensional polytopes where H-representations are infeasible. In such cases the number of vertices grows exponentially (e.g., a $K = 20$ dimensional hypercube has $2^{20} = 1048576$ vertices), which leads to numerical instabilities and computationally heavy quadratic programs. Thus, Aitchison flows represent an initial step for modeling polytopes in the V-representation. Our amortized variational approach contrasts with existing sampling-based methods in flux analysis (Theorell et al., 2024; 2017; Heinonen et al., 2019) and opens the door to fast simulation-based inference (Cranmer et al., 2020) and Bayesian optimal experiment design.

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

## A  SPLINE FLOWS ON THE BALL

Following the recursive construction in (Rezende et al., 2020), spline flows on the ball are obtained by mapping a point $s \in \mathbb{S}^K$ to cylinder coordinates via

$$\text{cyl}^D : \quad \mathbb{S}^D \times [-1,1]^{K-D} \to \mathbb{S}^{D-1} \times [-1,1]^{K-D+1},$$

$$c^D \mapsto c^{D-1} = \left[ \frac{c_{1:D-1}}{\sqrt{1-c_D^2}}, \, c_{D:K} \right]^T. \tag{48}$$

By composing these mappings from $D = K$ down to $D = 3$, we obtain an overall homeomorphism

$$\text{cyl} : \quad \mathbb{S}^K \to \mathbb{S}^1 \times [-1,1]^{K-1}, \quad s \mapsto c = \text{cyl}^3 \circ \cdots \circ \text{cyl}^K(s). \tag{49}$$

After applying $\text{cyl}^3$, the first two coordinates are converted into a single angle via

$$\theta = \text{atan2}(c_1, c_2). \tag{50}$$

This is the angle shown in figure 1. Together, these elements yield the composite coordinate vector

$$\varphi = [\theta, c_{3:K}, r], \tag{51}$$

where the radial component $r$ is derived from the hit-and-run ball transform. Distributions over $\theta$ are modeled using circular splines (Rezende et al., 2020) and conventional spline flows (Durkan et al., 2019a;b) are employed for the remaining dimensions. The resulting density for a point $v$ in the polytope is expressed as

$$q^{spline}(v) = p_\square^{\mathcal{U}}(\varphi) \cdot \left| \det \boldsymbol{J}^{spline} \right|^{-1} \cdot \left| \det \boldsymbol{J}^{v\varphi} \right|^{-1}, \tag{52}$$

where $\boldsymbol{J}^{spline}$ is the Jacobian of the flow parametrized by a neural network. $p_\square^{\mathcal{U}}(\varphi)$ is the uniform base-density on the hyper-rectangle $\square : [-\pi, \pi] \times [-1,1]^{K-2} \times [0,1]$ and the composite Jacobian $\boldsymbol{J}^{v\vartheta}$ captures all coordinate transforms from $\varphi$ to $v$. Vector $s \in \mathbb{S}^K$ expressed in cartesian coordinates has $K + 1$ elements and so does vector $c$. However, the full transformation $\text{atan2} \circ \text{cyl}^{3\cdots K} \circ \mathit{b}$ maps between $\mathcal{F}$ and $\square$, two $K$-dimensional manifolds, meaning that Jacobian $\boldsymbol{J}^{v\varphi}$ is square.

## B  TARGET DISTRIBUTION

We start from a small constraint-based metabolic model whose feasible fluxes form a convex polytope $\mathcal{F}$. The equality constraint matrix is

$$
\boldsymbol{S} = 
\begin{array}{c|ccccccccccccc}
 & c\_out & v1 & v2 & v3 & v4 & v5 & v6 & v7 & d\_out & f\_out & biomass & h\_out & a\_in \\
\hline
A & 0 & -1 & 0 & 0 & 0 & 0 & 0 & 0 & 0 & 0 & 0 & 0 & 1 \\
B & 0 & 1 & -1 & -1 & 0 & 0 & 0 & 0 & 0 & 0 & -0.6 & 0 & 0 \\
C & 0 & 0 & 0 & 1 & 0 & -1 & 0 & 0 & 0 & 0 & -0.1 & 0 & 0 \\
D & 0 & 0 & 0 & 1 & 1 & 0 & -1 & 0 & -1 & 0 & 0 & 0 & 0 \\
E & 0 & 1 & -1 & -1 & 1 & -1 & -1 & 0 & 0 & 0 & -0.5 & 0 & 0 \\
F & 0 & 0 & 0 & 1 & 1 & 0 & 1 & -2 & 0 & -1 & 0 & 0 & 0 \\
H & 0 & 0 & 0 & 0 & 0 & 0 & 0 & 1 & 0 & 0 & -0.3 & -1 & 0 \\
cof & -1 & 0 & 0 & 1 & 0 & 0 & 0 & 0 & 0 & 0 & 0 & 0 & 0 \\
\end{array}
\tag{53}
$$

As is typical in $^{13}$C-MFA, we set $h = 0$. The inequality constraints are $0.05 \le v_{\text{biomass}} \le 1.5$, $10 \le v_{a\_in} \le 10$ (an implicit equality), and $0 \le v_i \le 100$ for all remaining variables. We determine the minimal affine subspace via the RREF embedding (equation 12), which yields $K = 4$ with free variables $v7$, $h\_out$, $biomass$, and $f\_out$; the other nine fluxes are affine functions of these. We then apply the `PolyRound` pipeline (Theorell et al., 2022) to round the polytope to (approximate) John position (equation 19). In the figures and tables we denote these rounded independent coordinates by the $R\_$ prefix (e.g., $R\_v7$, $R\_f\_out$, $R\_biomass$, $R\_h\_out$). The numerical values of $(\boldsymbol{T}, \tau, \boldsymbol{E}, \epsilon)$ are provided in the accompanying Jupyter notebooks.

On this rounded $K = 4$ polytope we define our first target distribution, $p_{\mathcal{F}}^{mog}$, as a mixture of three Gaussians constrained to $\mathcal{F}$. Let the unconstrained mixture be

$$
p^{mog}(v) = \sum_{i=1}^{3} w^i \, \mathcal{N}(v; \mu^i, \boldsymbol{\Sigma}^i),
\tag{54}
$$

where $w^i$ are the mixture weights, $\mu^i$ the means, and $\boldsymbol{\Sigma}^i$ the covariance matrices. For $p_{\mathcal{F}}^{mog}$, we choose the means

$$
\begin{bmatrix}
 & R\_v7 & R\_f\_out & R\_biomass & R\_h\_out \\
\hline
\mu^1 = & -1.015 & 0 & 0 & 0 \\
\mu^2 = & 0 & 0 & 1.015 & 0 \\
\mu^3 = & 0 & 0 & 0 & 1.015 \\
\end{bmatrix}
\tag{55}
$$

and the mixture weights $(w_1, w_2, w_3) = \left(\frac{1}{4}, \frac{1}{4}, \frac{1}{2}\right)$ with covariances $\boldsymbol{\Sigma}^i = \frac{w_i}{8} \boldsymbol{I}$ for $i \in \{1, 2, 3\}$. The constrained density is

$$
p_{\mathcal{F}}^{mog}(v) = \frac{p^{mog}(v)}{\int_{\mathcal{F}} p^{mog}(u) \, du} = \frac{p^{mog}(v)}{Z_{\mathcal{F}}}.
\tag{56}
$$

Because $\mathcal{F}$ has dimensionality $K = 4$, we compute $\text{vol}(\mathcal{F})$ via `QHull` (Barber et al., 1996) and estimate

$$
Z_{\mathcal{F}} \approx \frac{\text{vol}(\mathcal{F})}{N} \sum_{i=1}^{N} p^{mog}(v_i), \qquad v_i \sim \mathcal{U}(\mathcal{F}), \quad N = 125{,}000,
\tag{57}
$$

yielding $Z_{\mathcal{F}} \approx 0.6336$. Because the inscribed unit ball of $\mathcal{F}$ touches every facet, choosing $\|\mu^i\|_2 \approx 1$ ensures that a substantial portion of the unconstrained mixture lies outside $\mathcal{F}$, which stresses support handling.

For the $K = 20$ hypercube $\square$ we define the second target $p_{\square}^{mog}$ by reusing the same weights $w^i$ and covariances $\boldsymbol{\Sigma}^i$; each $\mu^i$ has a single nonzero entry $1.015$ in one of the first three dimensions. After rounding, any hyper-cube will reduce to a hyper-cube with sides of length 2. The normalizing constant $Z_{\square}$ is computed analogously (with the hypercube volume available analytically) and is found to be $Z_{\square} \approx 0.1637$.

## C  Multi-proposal hit-and-run sampling of arbitrary densities over polytopes

Uniform sampling from polytopes is a topic that has been widely studied for decades; see for instance (Berbee et al., 1987; Lee & Vempala, 2022; Sun & Chen, 2024). For applications such as $^{13}$C-MFA, we are often interested in sampling non-uniform densities over a polytope. For instance, density $\pi$ could represent the posterior over fluxes, where at every proposal a labeling state needs to be simulated in order to compute the likelihood. The Elementary Metabolic Unit (EMU) algorithm (Antoniewicz et al., 2007) is one example of such a simulation algorithm. Labeling simulations generally consist of solving a cascade of linear systems, which can easily be parallelized on a GPU.

The number of density evaluations at every step of a sampling algorithm is $L \times M$, where $L$ is the number of chains and $M$ is the number of proposals (typically $M = 1$). Markov chain approaches are inherently sequential, and if the stationary distribution is complex, running more chains to increase parallelism might not speed things up, since individual chains need to converge (Geyer, 1992). For this reason, we developed the multi-proposal hit-and-run algorithm (Algorithm 1), where parallelism is increased by evaluating the density of multiple proposals ($M > 1$). In this publication, we are not dealing with labeling simulations for density evaluation but instead try to sample from a mixture of Gaussians constrained to a polytope. In this case too, our algorithm is a sensible choice since it allows for a tunable proposal distribution, the choice of which can significantly influence the convergence of the Markov chains.

---

**Algorithm 1:** Multiple proposal Hit-and-Run sampling of distributions with polytope support

---

**Input:** $\boldsymbol{A}, b$ defining a full-dimensional polytope $\mathcal{F} = \{v \in \mathbb{R}^K \mid \boldsymbol{A} \cdot v \leq b\}$
**Input:** $N$ number of samples in a single chain
**Input:** $M$ number of proposals to evaluate in a single step of the chain
**Input:** $\pi$ target density
**Input:** $q$ proposal density
**Output:** $\mathcal{Y}$ samples from (approx.) posterior

1  **function** chord_extremes*(v,s,$\boldsymbol{A}$, b)***:**
2   $\quad$ $d^s = \boldsymbol{A} \cdot s$
3   $\quad$ $d^v = b - \boldsymbol{A} \cdot v$
4   $\quad$ $\alpha = d^v \oslash d^s$
5   $\quad$ $\alpha^{min} = \max(\alpha \mid \alpha \leq 0)$
6   $\quad$ $\alpha^{max} = \min(\alpha \mid \alpha \geq 0)$
7   $\quad$ **return** $\alpha^{min}, \alpha^{max}$

8  **function** MCMC*($\boldsymbol{A}, b, N, M, \pi, q$)***:**
9   $\quad$ Sample initial point from ball: $v^0 \sim \mathcal{U}(\mathbb{B}^K)$
10  $\quad$ $\mathcal{Y} \leftarrow \{v^0\}$
11  $\quad$ $i \leftarrow 0$
12  $\quad$ **while** $i < N$ **do**
13   $\quad\quad$ Sample direction from sphere: $s \sim \mathcal{U}(\mathbb{S}^{K-1})$
14   $\quad\quad$ $\alpha^{min}, \alpha^{max} \leftarrow$ chord_extremes($v^0, s, \boldsymbol{A}, b$)
15   $\quad\quad$ $\alpha \leftarrow \left[\alpha_i \sim q(\alpha; \alpha^{min}, \alpha^{max}) \quad \forall i \in \{1, \ldots, M\}\right]^T$
16   $\quad\quad$ Proposals on the chord: $\mathcal{C} \leftarrow \{v^0\} \cup \{v^0 + s \cdot \alpha_i\}$
17   $\quad\quad$ Compute weights $w$ from equation 60 or equation 61 with proposals $\mathcal{C}$
18   $\quad\quad$ Accept proposal: $k \sim \text{Categorical}(w)$
19   $\quad\quad$ $v^0 \leftarrow \mathcal{C}_k, \mathcal{Y} \leftarrow \mathcal{Y} \cup \{\mathcal{C}_k\}, i \leftarrow i + 1$
20  $\quad$ **return** $\mathcal{Y}$

---

We base our multi-proposal MCMC on (Tjelmeland, 2004). When evaluating multiple proposals, there generally are two choices for transition kernels whose stationary distribution is density $\pi$. The first is the Barker transition kernel of equation 61, which is the one used in the original $M = 1$ Metropolis-Hastings algorithm (Metropolis et al., 1953). It is rarely seen in practice anymore, since (Peskun, 1973) proved that the asymptotic variance of estimators is lower when using the transition kernel of equation 60.

$$q(v^i \mid v^{/i}) = \prod_{j \in \{0:M\}, j \neq i} q(v^i \mid v^j) \tag{58}$$

$$= \prod_{j \in \{0:M\}, j \neq i} q(\alpha_i \mid \alpha_j) \tag{59}$$

$$\begin{cases} w_i(v^i) &= \frac{1}{M} \min\left(1, \frac{\pi(v^i) \cdot q(v^i \mid v^{/i})}{\pi(v^0) \cdot q(v^0 \mid v^{/0})}\right) \quad \forall i \in \{1, \ldots, M\} \\ w_0(v^0) &= 1 - \sum_{i \in \{1, \ldots, M\}} w_i \end{cases} \tag{60}$$

$$w_i = \frac{\pi(v^i) \cdot q(v^i \mid v^{/i})}{\sum_{i \in \{0:M\}} \pi(v^i) \cdot q(v^i \mid v^{/i})} \tag{61}$$

Note that for Algorithm 1 we do not sample proposals fully independently. Independent sampling would entail sampling a direction $s$ for every proposal, but this would increase code complexity since the computation of the terms in equation 59 would become more cumbersome. Also note that the proposal distribution is defined over scalar values $\alpha$. The two choices for proposal distribution are uniform: $q = \mathcal{U}(\alpha; \alpha^{\min}, \alpha^{\max})$ and truncated normal: $q = \mathcal{N}(\alpha; \alpha^{\min}, \alpha^{\max}, \mu = 0, \sigma^2)$. The truncated normal is centered on the current point (hence $\mu = 0$) and has a tunable parameter $\sigma^2$. Our algorithm allows for further tuning through the specification of a covariance matrix $\mathbf{\Sigma} \in \mathbb{R}^{K \times K}$. The variance along a chord can then be computed as follows: $\sigma^2 = s^T \cdot \mathbf{\Sigma} \cdot s$.

For the $p_{\mathcal{F}}^{mog}$ target density, we used a uniform proposal density with 3 proposals per step (excluding the current state) and adopted the Peskun transition kernel (equation 60). We ran 8 chains in parallel, discarding the first 1000 steps as burn-in and then thinning the chains by recording every 15th sample.

For the $p_{\mathcal{F}}^{unif}$ density, we configured the sampler with a uniform proposal density that generates a single proposal per step, combined with a Peskun transition kernel. This setup corresponds to the classical Metropolis-Hastings algorithm. Again, we ran 8 chains in parallel, with a 1000-step burn-in, and in this case every 10th sample was retained.

Convergence metrics for these samplings, including the $\hat{R}$ statistic and effective sample size (ESS), are summarized in Table 2. These metrics were computed using the `arviz` package (Kumar et al., 2019). We do not report the convergence statistics for the 20-dimensional $p_{\Box}^{mog}$ density here, but these details are available in the accompanying Jupyter notebooks.

| Target | Statistic | R_v7 | R_f_out | R_biomass | R_h_out |
|---|---|---|---|---|---|
| $p_{\mathcal{F}}^{mog}, S = 105k$ | ESS (%) | 19.9 | 56.3 | 16.6 | 14.7 |
| | $\hat{R}$ | 1.000312 | 1.000115 | 1.000488 | 1.000358 |
| $p_{\mathcal{F}}^{unif}, S = 125k$ | ESS (%) | 60.8 | 58.6 | 80.6 | 61.3 |
| | $\hat{R}$ | 1.000033 | 0.999992 | 1.000044 | 1.000013 |

Table 2: Convergence metrics for the MCMC sampling of a 4-dimensional polytope. ESS = effective sample size.

## D    EVALUATION METRICS

Let $\mathcal{S}$ denote a set of $S$ samples from a flow. We then compute the KL-divergence and effective sample size as follows:

$$w_i = \frac{p(x_i)}{q(x_i)} \quad \forall i \in \{1 : S\} \tag{62}$$

$$KL(q\|p) = \mathbb{E}_q[\ln q(x) - \ln p(x)] + \ln Z_{KL} \tag{63}$$

$$Z_{KL} = \mathbb{E}\Big[\frac{p(x)}{q(x)}\Big] \approx \frac{1}{S} \sum_{i=1}^{S} w_i \tag{64}$$

$$\text{ESS} = \frac{\sigma_{x\sim\mathcal{U}}^2[p(x)]}{\sigma^2\Big[\frac{p(x)}{q(x)}\Big]} \approx \frac{\Big(\sum_{i=1}^{S} w_i\Big)^2}{\sum_{i=1}^{S} w_i^2}. \tag{65}$$

Where $w_i$ are the importance weights, computed for target distribution $p$ and variational distribution $q$. The expectations in the formulas above are approximated by Monte-Carlo sampling.

## E  MODEL PARAMETERS

In this Section, we review the model architectures along with the training and inference performance of the flows employed in our experiments. The results are summarized in Table 3. The *div* column indicates the time required to generate 20k samples from the flow, including the computation of the log determinant or log divergence integral. Because CNFs incur additional overhead for evaluating the divergence integral, we also report the pure sampling time in the *sample* column.

For the $q^{spline}$ model, note that the *hid. dim.* is noted per transformation. For $q^{spline}$, the following architecture was used. We used a flow of 10 transformations interspersed with permutation layers. Each transformation is an auto-regressive rational quadratic spline flow (Durkan et al., 2019b) where the $\theta$ dimension was modeled as a circular spline flow (Rezende et al., 2020). Each spline had 30 bins (i.e., 31 knots, including end-points). We adapted the flows presented in the `normflows` package (Stimper et al., 2023). When sampling from $q^{spline}$, both the samples and log determinant are returned in one go, hence there being no value in the *sample* column. Although we did not perform a systematic hyperparameter search, our experiments indicate that reducing the complexity of the model (fewer hidden layers, fewer transformations, or lower hidden dimensions) degrades performance.

For inference with all CNFs, we used a midpoint numerical integrator with a step size of 0.05. In particular, for $q^{ball}$ we utilized the Riemannian version of the midpoint solver. Our CNF implementations and training procedures are based on adaptations of the algorithms presented in the `flow_matching` package (Lipman et al., 2024). We computed the divergence integral using automatic differentiation to ensure accurate estimates, and we implemented the forward divergence (integrating from $t = 0$ to $t = 1$) estimation ourselves, as this functionality was not yet available in the package.

All experiments were performed on a laptop with an Intel(R) Core(TM) i7-7700HQ @ 2.80GHz CPU and an NVIDIA GeForce GTX 1060 6GB GPU, which is CUDA enabled and was utilized for all experiments.

## F  LLM USE

The paper was written by the first author. To improve the flow of the paper, some sections of text were revised using guidance from LLMs (ChatGPT4 and ChatGPT5). The math was written by hand, with the occasional refactor of symbols by an LLM.

| Model | target | dim | hid. lay. | hid. dim. | lr | epochs | batch size | train (s) | div (s) | sample (s) |
|---|---|---|---|---|---|---|---|---|---|---|
| $q^{spline}$ | $p_{\mathcal{F}}^{mog}$ | 4 | 4 | 64 | 4e-3 | 35 | 12288 | 1504 | 1.5 | - |
| $q^{eucl}$ | $p_{\mathcal{F}}^{mog}$ | 4 | 6 | 512 | 1e-3 | 35 | 8192 | 200 | 74 | 3.8 |
| $q^{ball}$ | $p_{\mathcal{F}}^{mog}$ | 4 | 6 | 512 | 1e-3 | 35 | 8192 | 247 | 32 | 1.6 |
| $q^{ait}$ | $p_{\mathcal{F}}^{mog}$ | 4 | 6 | 512 | 1e-3 | 50 | 8192 | 213 | 12 | 1.7 |
| $q^{eucl}$ | $p_{\square}^{mog}$ | 20 | 6 | 1024 | 1e-3 | 35 | 8192 | 302 | 998 | 11.4 |

Table 3: Comparison of key architectural parameters and corresponding training and inference times for various models. The *sample* column denotes the time to sample from the flow, whereas the *div* column denotes the time necessary to track the divergence of the CNF, which is necessary to track the density of a sample.

