# OpenReview forum: "Flows on convex polytopes"
_ICLR.cc/2026/Conference — Submitted to ICLR 2026_

### Official Review · Reviewer_7twp · 2025-10-29

**Soundness:** 2
**Presentation:** 1
**Contribution:** 2
**Rating:** 2
**Confidence:** 2

**Summary:**

This paper aims at modeling distributions on convex polytopes using the flow matching framework. The proposed idea is to
- map the convex polytope to a unit ball
- use the existing flow matching framework on Riemannian manifold to model the distribution
- then map them back to the original polytope.

Such a strategy is tested on modeling the mixture of Gaussians with some convex constraints.

**Strengths:**

This paper proposed a working-around strategy to model some constrained distributions using existing riemannian flow matching.

**Weaknesses:**

- The presentation is poor, and the paper is full of equations (nearly 50 equations) without elaborating their roles.
- The experiment remains on a test level without proper setup and applications.

**Questions:**

no other specific questions.

---

### Official Review · Reviewer_pjrk · 2025-10-31

**Soundness:** 3
**Presentation:** 2
**Contribution:** 2
**Rating:** 2
**Confidence:** 4

**Summary:**

This paper proposes a framework for generative modeling of distributions defined on convex polytopes. The authors introduce two strategies depending on the polytope's representation. The first, designed for polytopes with half-space (H) representation, maps the polytope to the unit ball and then applies current generative models, such as continuous normalizing flows (CNFs) or spline flows, to the simpler ball geometry.

The second strategy is designed for polytopes with a vertex (V) representation. While it is possible to convert to the H representation, it is computationally expensive and therefore the authors dedicate a seperate algorithm for this instance. This method first maps points from the polytope to unique barycentric coordinates on a simplex using maximum entropy coordinates (mec). It then applies an isometric log-ratio (ilr) transform to map these coordinates to a K-dimensional Euclidean space, where a standard Euclidean CNF is trained.

The authors evaluate these methods on synthetic mixture-of-Gaussian distributions constrained to 4D and 20D polytopes, comparing their performance against a baseline Euclidean CNF.

**Strengths:**

The paper presents an interesting and potentially valuable framework for a specific problem domain where generative modeling on polytopes is the main focus, such as certain classes of optimization problems. The methodological derivations appear sound. The authors are careful to use smooth, bijective transformations (diffeomorphisms) at each step of their proposed mappings. This ensures that log-likelihoods remain well-defined, allowing the framework to be compatible with standard normalizing flow training and evaluation, even if it is not strictly necessary for flow matching.

**Weaknesses:**

The paper is broadly unclear for a general machine learning audience. It assumes significant expertise in polytope geometry and its associated algebra. Critical concepts, such as the H-representation (Equations 1-4), "John polytope," "maximum entropy coordinates," and "Aitchison geometry", are introduced with little to no explanation, making the paper difficult to follow for non-experts. These foundational concepts need to be introduced more comprehensively.

Beyond the transformations to the unit ball and Euclidean space (whose originality is not fully clarified), the core generative modeling methods are applications of existing techniques (Flow Matching, normalizing flows). The primary contribution appears to be the careful composition of these transformations and the tracking of their Jacobians, which is mainly relevant for log-likelihood evaluation, but may not be critical for the training of the CNF models (like Flow Matching) themselves.

**Questions:**

The paper fails to make a strong case for the real-world importance of generative modeling on polytopes. All experiments are conducted on simulated data (mixtures of Gaussians). While a Euclidean flow baseline is included (and shown to be insufficient), the lack of comparisons to other potential baselines or any application to a real-world problem makes it difficult to judge the practical value of the proposed framework. A clearer demonstration of why the "naive" approach of projecting samples is computationally inferior and an application to a problem from metabolic modeling (as hinted at), would have significantly strengthened the paper.

---

### Official Review · Reviewer_hZ5o · 2025-10-31

**Soundness:** 3
**Presentation:** 3
**Contribution:** 2
**Rating:** 4
**Confidence:** 3

**Summary:**

The paper introduces a novel framework for applying normalizing flows on convex polytopes. Motivated by recent advanced in Riemannian flow matching and normalizing flows, the authors make the connection to applications in areas that involve convex optimization of systems to produce simulations of samples. They then derive a framework to perform invertible transformations to allow for training of discrete and continuous normalizing flows on the convex polytope. They evaluate their method on single example from the field of metabolic modeling. The paper proposes intersting methods but falls short in experimental evaluation.

**Strengths:**

The authors provide a clear motivation and derivation of their method. Sections 2 and 3 are excellent in their flow of ideas, introducing the domain of convex optimization on polytopes to their relation to normalizing flows on Riemannian manifolds. I think they could be improved by making a table that summarizes the pros/cons of the different flows that are then evaluated in the experiments.

**Weaknesses:**

The paper's weaknesses lie mainly in thoroughness of evaluation. Only two examples from a simple mixture of Gaussians model are used. I would hope to see an evaluation on data, for example, from a metabolic simulator. Similar to what I said in the strengths, I think the methods or experiments section could use a table that highlights the tradeoffs of the various flows that were evaluated. I find it strange that the euclidean flow performs just as well as the other flows and wonder if that combined with rejection sampling is sufficient, obviating the need for Riemannian or ball flows. Additionally, the authors had to derive manual jacobians for the transforms which limits the scalability of the proposed flows.

Here are other standalone critiques:
-  typo in lines 293-294 and line 359

**Questions:**

I included some questions in the weaknesses and add more standalone questions here:
- Flows are normalized by construction, so they shouldn't need to be divided by a normalizing ocnstant... and also how are they getting that normalizing constant?
- Why not include the boundary of the ball in Riemannian flow matching? That seems sensible unless there's a technical challenge? I think it's because of the maximum entropy coordinates but I'm not completely sure.
-  What are the consequences for choosing MEC to find barycentric coordinates? I'm assuming there are other formulations. Kind of like how they could've used the Fisher-Rao metric, which would be interesting to know why they chose Aitchison over Fisher-Rao very briefly.
- Looks like their model isn't really handling the edges as well as the MCMC sampler in Figure 3. Why is that and how can it be addressed?
- They mention BOED but it isn't well-motivated here and is missing a reference to the seminal Lindley paper.
- Why not perform simulation based inference with these flows? It seems as though that would be helpful experiments to evaluate their method. They could compare the different efficiencies of the different flow models in 1) creating posterior distributions and 2) evaluating calibration of the posteriors. Their method doesn't seem to be too far off and I think that would _greatly_ improve confidence in the method. In that case, I would feel more confident in their method and the tradeoffs of different flows from those evaluations.

---

### Official Review · Reviewer_NsM4 · 2025-11-01

**Soundness:** 3
**Presentation:** 3
**Contribution:** 3
**Rating:** 6
**Confidence:** 2

**Summary:**

The authors demonstrate differentiable mappings from a convex polytope to a unit ball (for both H and V representations) which allow for the application of both normalising flow and flow matching models to model distributions on the unit ball which is much more stable and easier than learning a flow on the convex polytope directly. This is then evaluated empirically on a range on problems.

**Strengths:**

* The approach appears to well formulated from mapping a convex polytope to a unit ball (for either the H or V representation) to allow for the established machinery of flows on the unit ball to be applied for generative modelling.

**Weaknesses:**

* I'm not sure how well the machine learning community is familiar with convex polytopes. This makes the paper a bit hard to follow. It would be good to provide further details on this area in the appendix.

* Novelty appears to be slightly limited as it seems that the mapping from a convex polytope to a unit ball is already well known in the literature and generative models on the unit ball is also well established. The main novelty is the application of both to the useful problem of modelling distributions on convex polytopes.

**Questions:**

* Line 67: "equation 1 shows" should be capitalised.

---

### Meta-Review · Area_Chair_iZVW · 2026-01-07

**Summary:**

The paper addresses the challenge of generative modeling (learning and sampling distributions) on convex polytopes—geometric shapes defined by flat faces and vertices (e.g., a cube or a complex metabolic flux space).  The authors propose two primary strategies to map these constrained spaces into simpler geometries where standard Generative Models (like Normalizing Flows or Flow Matching) can operate.

**Reviewer Concerns:**

Multiple reviewers (NsM4, pjrk, 7twp) noted that the paper is "hard to follow" for a general machine learning audience. It relies heavily on advanced geometry (John polytopes, Aitchison geometry, Jacobians) without providing enough introductory context.

Specifically, reviewer 7twp critiqued the paper for having nearly 50 equations without sufficient explanation of their roles.

**Reviewer Scores:**

The authors remained inactive during the rebuttal and discussion period. I reckon the reviewers won't have any motivation to change the score.

---

### Decision · Program_Chairs · 2026-01-26

Reject